# Community burden of hypertension and treatment patterns: An in-depth age predictor analysis: (The Rural Community Risk of Non-Communicable Disease Study - Nyive Phase I)

**James Osei-Yeboah**[1]*, **Ellis Owusu-Dabo**[1], **William K. B. A. Owiredu**[2,3], **Sylvester Yao Lokpo**[4], **Francis Delali Agode**[4,5], **Beatrice Bella Johnson**[6]

1 School of Public Health, Kwame Nkrumah University of Science and Technology, Kumasi, Ghana, 2 Department of Molecular Medicine, School of Medical Sciences, Kwame Nkrumah University of Science and Technology, Kumasi, Ghana, 3 Department of Clinical Biochemistry, Diagnostic Directorate, Komfo Anokye Teaching Hospital, Kumasi, Ghana, 4 Department of Medical Laboratory Sciences, School of Allied Health Sciences, University of Health and Allied Sciences, Ho, Ghana, 5 Laboratory Department, Akatsi South District Hospital, Akatsi, Ghana, 6 Department of Nursing, School of Nursing and Midwifery, University of Health and Allied Sciences, Ho, Ghana

* jimjoy223@gmail.com

**Data Availability Statement:** All relevant data are in the paper and its Supporting Information files.

## Abstract

### Background

This study aimed to describe the burden, treatment patterns and, age threshold for predicting hypertension among rural adults in Nyive in the Ho Municipality of the Volta Region, Ghana.

### Methods

A population-based cross-sectional study design was employed. A total of 417 adults aged 20 years and above were randomly selected from households within the Nyive community. The WHO STEPwise approach for non-communicable diseases risk factor surveillance (STEPS) instrument was used to obtain socio-demographic and clinical information including age, gender, educational background, marital status, and occupation as well as hypertension treatment information. Blood pressure was measured using standard methods. The risk of hypertension and the critical age at risk of hypertension was determined using binary logistic regression model and the receiver-operator characteristics (ROC) analysis.

### Results

The direct and indirect age-standardized hypertension prevalence was higher in males (562.58/487.34 per 1000 residents) compared to the females (489.42/402.36 per 1000 residents). The risk of hypertension among the study population increased by 4.4% (2.9%-5.9% at 95% CI) for one year increase in age while the critical age at risk of hypertension was >39 years among females and >35 years among males. About 64(46.72%) of the hypertensive participants were not on treatment whereas only 42(30.66%) had their blood pressure controlled.

**Funding:** The author(s) received no specific funding for this work.

**Competing interests:** The authors have declared that no competing interests exist.

## Conclusion

Rural hypertension is high among adults in Nyive. The critical age at risk of hypertension was lower among males. The estimated annual increase of risk of hypertension was 4.7% for females and 3.1% for males. High levels of undiagnosed and non-treatment of hypertension and low levels of blood pressure control exist among the rural folks.

## 1. Background

Hypertension is a major risk contributor to stroke, ischaemic heart disease, and kidney failures [1, 2] as well as premature death [3]. Current reports by the World Health Organization (WHO) indicate that an estimated 1.13 billion people have hypertension globally, with two-thirds of the affected people living in low- and middle-income countries [4]. Of all the WHO Regions, sub-Saharan Africa is reported to have the highest prevalence of hypertension (27%) [4]. In Ghana, varied reports of hypertension prevalence have been published, with rates ranging from 22–44.7% over the past years [5–8]. Heavy alcohol consumption [9] overweight and obesity [6], as well as sedentary lifestyles [10] are some factors postulated to contribute to the rising burden of hypertension in sub-Saharan Africa. Moreover, previous studies have suggested a higher burden of hypertension among Ghanaian urban dwellers compared to their rural folks. Over the years, however, the gap in the rates between the two settings appeared to have narrowed. For example, a study conducted by Cappuccio, Micah [7] in the year 2004 revealed hypertension rates among rural and urban dwellers in the Ashanti Region to be 24% and 32%, respectively. While in 2017, Solomon, Adjuik [10] reported rates of 34% and 36% among rural and urban dwellers respectively in the Hohoe Municipality. The results could also indicate a phenomenon of a rising hypertension trend among Ghanaians in rural communities, as reported earlier by Cook-Huynh, Ansong [11]. Besides, earlier studies have suggested an early onset of hypertension among the Ghanaian populace [12, 13].

The Community-Based Health Planning and Services (CHPS) policy is a national strategy rolled out in Ghana to provide primary healthcare services to deprived communities including those in rural areas [14] after a successful pilot study in Navrongo located in Northern Ghana [15]. Since then, studies have explored the possibility of incorporating hypertension care into the CHPS program in a strategy dubbed "the nurse-led task-shifting strategy for hypertension control" (TASSH) yielding a relative success in blood pressure control when combined with the health insurance coverage initiative [16, 17]. However, financial, logistical, and telecommunication challenges, lack of recognition and cooperation from community members, lack of motivation, and lack of regular skill development training programs for community health management committees (CHMC) remain barriers to the successful implementation of the CHPS program [18]. Moreover, there is also limited data on the hypertension burden and control in most rural areas of sub-Saharan African countries including Ghana [19]. It is against this backdrop that we designed the current study to investigate the prevalence of hypertension, treatment patterns, and predictive age thresholds for diagnosing hypertension among rural adults in the Nyive community in the Ho Municipality of the Volta Region, Ghana.

## 2. Materials and methods

### 2.1 Study area and study site description

Nyive is a rural community found on the left bank of River Tordzie in the northern part of the Ho Municipality. The Nyive community has four CHPS zones providing primary health

services to the community. The Municipality has Ho as its capital and also serves as the capital and economic hub of the Volta Region. The Municipality is located between latitudes 6°20"N and 6°55"N and longitudes 0°12'E and 0°53'E. The Municipality shares boundaries with Adaklu and Agotime-Ziope Districts to the South, Ho West District to the North and West, and the Republic of Togo to the East. Its total land area is 2,361 square kilometers thus representing 11.5 percent of the region's total land area. The population of the district according to the 2010 population and housing census was 177,281, with 83,819 being males and 93,469 females. A total of 110,048 of the population, representing 62.1% live in urban areas while 37.9% live in rural areas.

## 2.2 Study design, study population, and sampling technique

A population-based cross-sectional study was conducted from August to September 2018. A total population of 3,110 was projected in Nyive based on the enumerated data from the Ghana Universal Long-Lasting Insecticidal Net (LLIN) distribution in a pre-distribution household data validation for 2018. The national LLINs distribution program was carried out between May 2010 and October 2012 in Ghana where about 12.5 million nets were distributed to households. The program involved pre-registration of households and their sleeping places, door-to-door distribution of LLINs by volunteers, and post-distribution behaviour change communication activities to encourage high and sustained use [20]. A projected eligible population of 1,840 with 999 females and 841 males (20 years and above) was estimated using the age and sex distribution from the 2010 population and housing census for the area. Out of a total of 425 eligible adults aged 20 years and above, 417 consented to participate in this study constituting an acceptance rate of 98.12%. Participants who were randomly selected from households based on the population density of the four CHPS zones are part of an ongoing study of the Rural Community Risk of Non-Communicable Disease Study-(Nyive Phase I) Cohort. The Rural Community Risk of Non-Communicable Disease Study is a baseline study aimed at understanding the burden of non-communicable diseases including type 2 diabetes, hypertension, and co-morbid conditions as well as the risk drivers within a rural context using Nyive as the study community in the Ho Municipality of the Volta Region.

## 2.3 Data collection techniques and tools

The WHO STEPwise approach for non-communicable diseases risk factor surveillance (NCD) instrument was used to collect data for this study. In brief, the instrument covers three different levels or 'steps' of risk factor assessment: Step 1 (demographic information), Step 2 (physical measurements), and Step 3 (biochemical measurements). Step 1 captures information related to socio-demography (age, gender, educational level, marital status, employment status, income) and behavourial or lifestyle parameters (tobacco use, alcohol consumption, dietary characteristics, physical activity, history raised blood pressure, and diabetes). Step 2 captures information related to physical measurements (weight, height, waist circumference, blood pressure, hip circumference, and heart rate). Step 3 captures information related to biochemical measurements (blood glucose, blood lipids (total cholesterol), triglyceride, and high-density lipoprotein cholesterol) [21].

## 2.4 Blood pressure measurements

After resting for 3–5 minutes, blood pressure was measured in the non-dominant arm using fully automated blood pressure monitor (OMRON Healthcare, Intelli-sense BP785, HEM-7222, USA) in the sitting position and at arm's level during the time they were fasting 10–12

hours. A qualified health practitioner performed the blood pressure measurements and the average of three consecutive readings taken about 2 minutes apart was recorded.

## 2.5 Definition of hypertension and haemodynamic presentations

Hypertension was defined as a self-report of a previous diagnosis of hypertension and/or being on antihypertensive medication. Haemodynamic presentations among the undiagnosed participants were assessed using the Seventh Report of the Joint National Committee (JNC VII) on Prevention, Detection, Evaluation, and Treatment of High Blood Pressure criteria. This criterion remain the hypertension diagnosis guidelines in used the jurisdiction, as stated in the 7th edition of the Standard Treatment Guidelines of the Ghana Ministry of Health [22]. Normotensives were classified as systolic blood pressure (SBP) < 120 mmHg and diastolic blood pressure (DBP) < 80 mmHg, prehypertension (SBP 120–129 mmHg or DBP 80–89 mmHg), hypertension stage 1 (SBP 140–159 mmHg or DBP 90–99 mmHg), and hypertension stage 2 (SBP ≥ 160 mmHg or DBP ≥ 100 mmHg) [23]. Age at diagnosis was defined as the age of participants at the time of diagnosis for self-reported or previously diagnosed participants and the current age for newly diagnosed participants.

## 2.6 Sample size calculation

A minimum sample size of 318 was calculated from the community's expected population of 1,840, at a 95% confidence interval, an acceptable margin of error of 5%, and a response rate of 50%. The Online Raosoft sample size calculator was used (www.raosoft.com.14). A total of 417 participants were used in this study.

## 2.7 Statistical analysis

A continuous variable was expressed as mean ± standard deviation and categorical variables were expressed as frequency and proportion. The difference between proportions was tested with Fisher exact and at all times an alpha of less than 0.05 was considered statistically significant. Crude incidence and age-standardized incidence were calculated using data from the 2010 Population and Housing Census for the study area and 2018 LLIN pre-distribution household data validation as the base population. Population weights were calculated as the total number of people who fell within a specific age group (n) divided by the total eligible population (N). A binary logistic model was used to assess the predictability of hypertension by age. The Youden index was computed to identify population-specific age cut-off points for the optimal differentiation between hypertension and non-hypertension. The Youden Index was derived from (sensitivity + specificity) - 1. Using the area under the curve (AUC) of a receiver-operator characteristic curve (ROC), the gender-specific predictive threshold for identifying hypertension cases (discrimination) was estimated. IBM Statistical Package for the Social Sciences version 22.00 (SPSS Inc, Chicago, USA (http://www.spss.com)) and MedCalc version 12.3.2 for windows (MedCalc Software bvba, Acacialaan 22, B-8400 Ostend, Belgium, (www.medcalc.org) were used for data analysis.

## 2.8 Ethical consideration

Ethical clearance was sought, and ethical approval was given by the Ethical and Protocol Review Committee of the University of Health and Allied Sciences, Ho, Ghana with protocol number **UHAS-REC A.4 [192] 18–19**. Signed written informed consent was obtained from the participants who could read and write. The aim and processes of the research were fully

explained to the participants who could not read, and a thumbprint was obtained. Participation was voluntary and confidentiality of data was guaranteed.

## 3. Results

The average age of the respondents was 50.62 ranging from a minimum of 20 years to a maximum of 85 years. Per the age categorization, 44.13% of the respondents were below 50 years. The proportion of female participants was higher (81.29%). The majority of the participants (74.59%) had attained not more than basic education at the time of the survey. Most of the respondents had ever married and were employed in the informal sector ([Table 1]).

Out of the 417 participants, 402 representing 96.40% attested ever having their blood pressure measured for hypertension. Among those ever tested, 32.85% affirmed been diagnosed with hypertension. Among these known hypertensive participants, 64(46.72%) were not on treatment, 73(53.28%) had ever taken medication for hypertension (orthodox, herbal or both), 38.69% had ever consulted a traditional healer on hypertension and 19.71% were on herbal medication for the management of hypertension. While 12.41% combined both orthodox and herbal medications for hypertension management, 7.30% and 33.58% either used herbal

**Table 1. Demographic characteristics of residents of the Nyive community.**

| Parameter | Frequency | Percentage |
|---|---|---|
| Total | 417 | 100 |
| **Age of Participants (years)** | | |
| Mean Age | 50.62±15.39* | (20–85)* |
| 20–29 | 40 | 9.59 |
| 30–39 | 71 | 17.03 |
| 40–49 | 73 | 17.51 |
| 50–59 | 108 | 25.90 |
| 60–69 | 63 | 15.11 |
| 70–79 | 47 | 11.27 |
| 80–89 | 15 | 3.60 |
| **Gender** | | |
| Female | 339 | 81.29 |
| Male | 78 | 18.71 |
| **Educational Background** | | |
| None | 77 | 18.47 |
| Basic | 234 | 56.12 |
| Secondary | 79 | 18.94 |
| Tertiary | 27 | 6.47 |
| **Marital Status** | | |
| Single | 36 | 8.63 |
| Married | 254 | 60.91 |
| Divorce | 54 | 12.95 |
| Widowed | 73 | 17.51 |
| **Occupation** | | |
| None | 12 | 2.88 |
| Informal | 381 | 91.37 |
| Formal | 24 | 5.76 |

Data is presented as frequency and proportion.

* Data is presents as mean±standard deviation with minimum and maximum in parenthesis.

**Table 2. Prevalence of known hypertension and treatment options among residents of Nyive community.**

| Parameter | Total | Female | Male | p value |
|---|---|---|---|---|
| Ever had BP Measured | 402(96.40) | 331(97.64) | 71(91.03) | 0.011 |
| Diagnosed of Hypertension | 137(32.85) | 117(34.51) | 20(25.64) | 0.143 |
| Diagnosed of Hypertension within the last 12 months | 92(67.15) | 78(66.67) | 14(70.00) | 1 |
| Controlled Blood Pressure | 42(30.66) | 35(29.91) | 7(35.00) | 0.793 |
| *Medication Profile* | | | | |
| Not on Treatment | 64(46.72) | 53(45.30) | 11(55.00) | |
| Hypertension Medication within 2 Weeks | 63(45.99) | 55(47.01) | 8(40.00) | 0.632 |
| Consulted Traditional Healer on Hypertension | 53(38.69) | 47(40.17) | 6(30.00) | 0.463 |
| Herbal Medication for Hypertension | 27(19.71) | 25(21.37) | 2(10.00) | 0.364 |
| Herbal Medication only | 10(7.30) | 9(7.69) | 1(5.00) | |
| Orthodox Medication only | 46(33.58) | 39(33.33) | 7(35.00) | |
| Both Herbal & Orthodox | 17(12.41) | 16(13.68) | 1(5.00) | |

Data are presented as the frequency with the corresponding percentage in parenthesis. p is significant at 0.05. BP- Blood pressure.

medication and orthodox medication for hypertension management respectively. The level of controlled blood pressure recorded among known hypertension was 30.66% overall, 29.91% among the female subgroup, and 35.00% among the males (Table 2).

Among the participants who had not been diagnosed with hypertension, 75 out of 280 were diagnosed of having high blood pressure. The percentage burden of newly diagnosed hypertension was significantly higher among the male subpopulation (41.38%) compared to their female counterparts (22.79%). Among the newly diagnosed hypertensive participants, 34.67% were classified as stage two hypertension, 46.67% presented with isolated high systolic blood pressure (SBP), 26.67% with isolated high diastolic blood pressure, and the rest with both high SBP and DBP (26.67%) (Table 3).

The crude prevalence of hypertension among the study population was estimated at 508.39 per 1000 residents, 495.58 per 1000 female, and 564.10 per 1000 male subpopulations respectively. Upon a direct age-adjustment using the total surveyed population, the direct age standardized prevalence of hypertension was estimated at 489.42 and 562.58 per 1000 residents among the female and the male subpopulations respectively. After an indirect age adjustment using the actual population of the study area in 2018, the age-standardized prevalence of Nyive was estimated at 402.36 per 1000 female residents for the female group and 487.34 per 1000 male residents for the male group. In general, the prevalence of age group-specific hypertension recorded a rise from 20 to 49 years before a decline with older year groups (Table 4).

Using a population pyramid of hypertension over a single year of the age distribution for the general population, a distribution with a heavy base cluster with a mean age of 45.83±15.62 and mode of 46.00 was observed for the non-hypertension population. On the other hand, a flat base distribution with a heavy cluster from the middle to the top at a mean age of 55.26 ±13.69 and a modal mark of 55.00 was observed for the hypertension population. A significant average age difference tilted towards the hypertension population was observed (p <0.0001) (Fig 1).

The non-hypertension female group was found to cluster at the lower base of the pyramid leaving a flat top with a median age of 46 and an age mean of 45.87±14.98 which was significantly lower (p<0.0001) compared to an age mean of 55.21±13.25 for the hypertension group. The hypertension group was found to cluster heavily from the middle (≥40) to the top of the pyramid with a flat base (Fig 2).

**Table 3. Prevalence of undiagnosed hypertension and haemodynamic presentation of community residents in Nyive.**

| Parameter | Total (280) | Female (222) | Male (58) | p value |
|---|---|---|---|---|
| **Normotensive** | 109(38.93) | 91(40.99) | 18(31.03) | |
| (SBP<120 and DBP<80) | | | | |
| **Prehypertension** | 96(34.29) | 80(36.04) | 16(27.59) | |
| (SBP 120–139 or DBP 80–89) | | | | |
| **Hypertension** | 75(26.79) | 51(22.97) | 24(41.38) | 0.0019 |
| (SBP ≥140 or DBP ≥90) | | | | |
| *Stages of Hypertension* | | | | |
| **Stage 1 Hypertension** | 49(65.33) | 31(60.78) | 18(75.00) | 0.3011 |
| (SBP 140–159 or DBP 90–99) | | | | |
| **Stage 2 Hypertension** | 26(34.67) | 20(39.22) | 6(25.00) | |
| (SBP ≥160 or DBP ≥100) | | | | |
| *Pattern of Hypertension* | | | | |
| **Isolated SBP Hypertension** | 35(46.67) | 26(50.98) | 9(37.50) | 0.2621 |
| (SBP ≥140 and DBP < 90) | | | | |
| **Isolated DBP Hypertension** | 20(26.67) | 11(21.57) | 9(37.50) | 0.0052 |
| (SBP < 140 and DBP ≥90) | | | | |
| **Both SBP & DBP Hypertension** | 20(26.67) | 14(27.45) | 6(25.00) | 0.2341 |
| (SBP ≥140 and DBP ≥90) | | | | |

Data are presented as the frequency with the corresponding percentage in parenthesis. p is significant at 0.05. SBP-Systolic blood pressure, DBP-Diastolic blood pressure.

In the male subpopulation, the non-hypertension group was distributed around a population mean of 45.65±18.74, significantly lower (p = 0.0133) compared to the age mean of 55.45 ±15.48 for the hypertension group. As seen in Fig 3, the population density for the non-hypertension group was heaviest below 40 years while the population density bulges after 40 years among the hypertension group (Fig 3).

Using a binary logistic model adjusted for education, it was observed that, for a one year increase in age, the risk of hypertension among the population increases by 4.4% (2.9%-5.9% at 95% CI), this odds was 4.7% (3.0%-6.5% at 95% CI), among the female subpopulation and 3.1% (0.2%-6.0% at 95% CI), among the male subpopulation. The model correctly classified

**Table 4. Crude and age standardized prevalence rate of hypertension among dwellers of the Nyive community.**

| Parameter | Crude Rate / 1000 | | Direct Age Standardized Rate/1000 | | | Weight Community* | | Indirect Age Standardized Rate / 1000 | |
|---|---|---|---|---|---|---|---|---|---|
| Age Group | Female | Male | Survey Weight* | Female | Male | Female | Male | Female | Male |
| **20–29 years** | 71.43 | 333.33 | 0.1 | 6.85 | 31.97 | 0.26 | 0.28 | 18.45 | 92.75 |
| **30–39 years** | 288.14 | 250 | 0.17 | 49.06 | 42.57 | 0.2 | 0.21 | 57.11 | 52.02 |
| **40–49 years** | 569.23 | 750 | 0.18 | 99.65 | 131.29 | 0.19 | 0.18 | 105.98 | 132.88 |
| **50–59 years** | 566.67 | 666.67 | 0.26 | 146.76 | 172.66 | 0.14 | 0.14 | 81.11 | 92.75 |
| **60–69 years** | 574.47 | 750 | 0.15 | 86.79 | 113.31 | 0.09 | 0.09 | 54.05 | 69.56 |
| **70–79 years** | 690.48 | 400 | 0.11 | 77.82 | 45.08 | 0.09 | 0.07 | 58.75 | 26.16 |
| **80–89 years** | 625 | 714.29 | 0.04 | 22.48 | 25.69 | 0.04 | 0.03 | 26.9 | 21.23 |
| **Total** | **495.58** | **564.10** | 1 | **489.42** | **562.58** | 1 | 1 | **402.36** | **487.34** |

*Survey Weight: Based on the total eligible population.

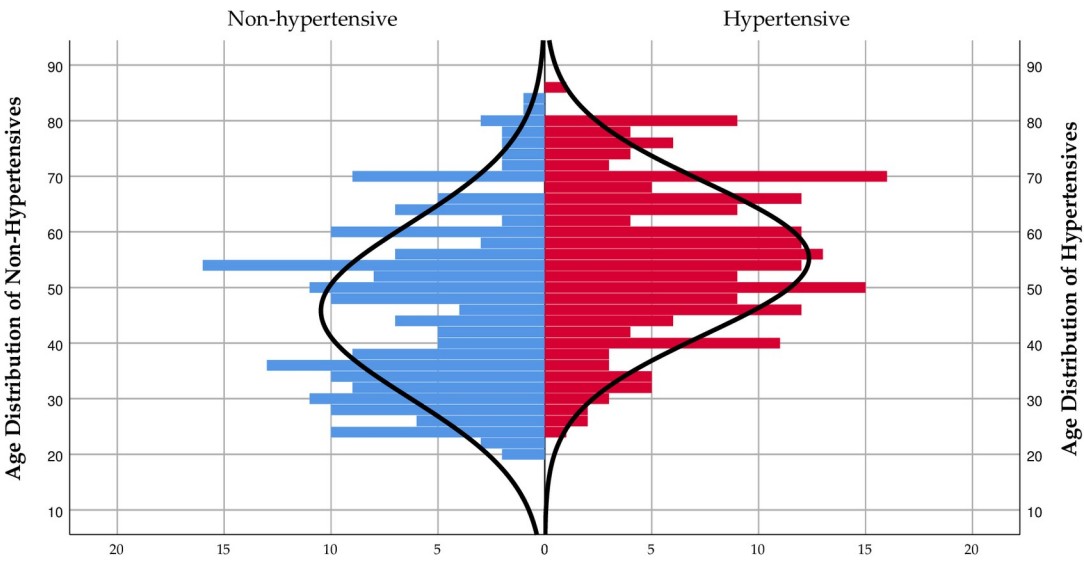

**Fig 1. Age distribution population pyramid over hypertension among residents of Nyive.**

60.67% of the population, 60.49% of the non-hypertension population, and 60.85% of the hypertension population.

As seen in Table 5, the critical age threshold of >39 years with a discriminating power of 0.679 for hypertension was observed among the female subpopulation using receiver operating characteristics curve analysis. In the male subpopulation, the critical at-risk age for hypertension (i.e. the age cutoff at which a person have a risk of developing hypertension) was >35 with a discriminating power of 0.658, a sensitivity of 86.36, and specificity of 50.00 (Table 5).

## 4. Discussion

Hypertension contributes to a major disease burden in Ghanaian adults, with variable rates documented among urban and rural dwellers [5, 8]. To date, there is no comprehensive

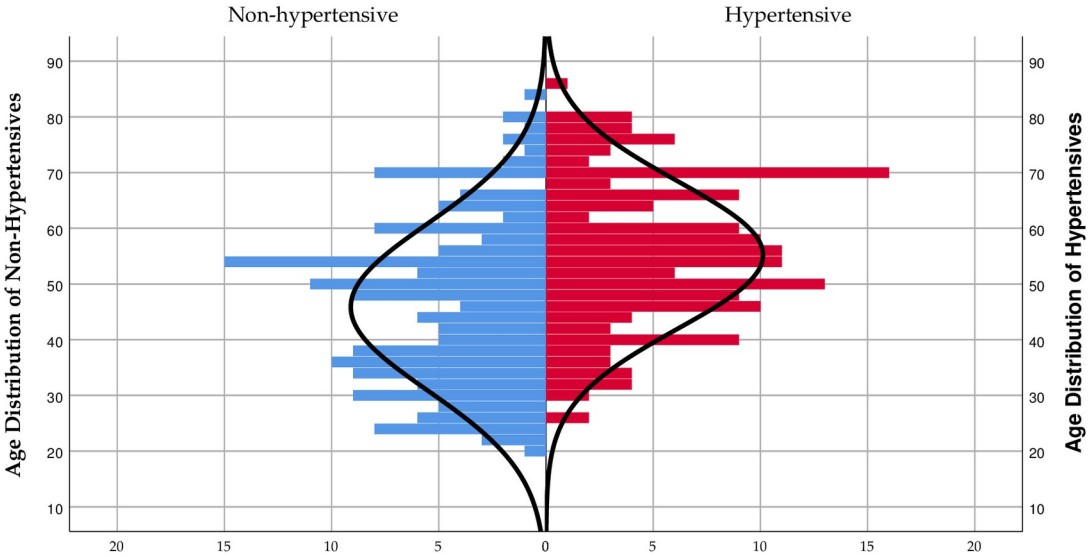

**Fig 2. Age distribution population pyramid over hypertension among female residents of Nyive.**

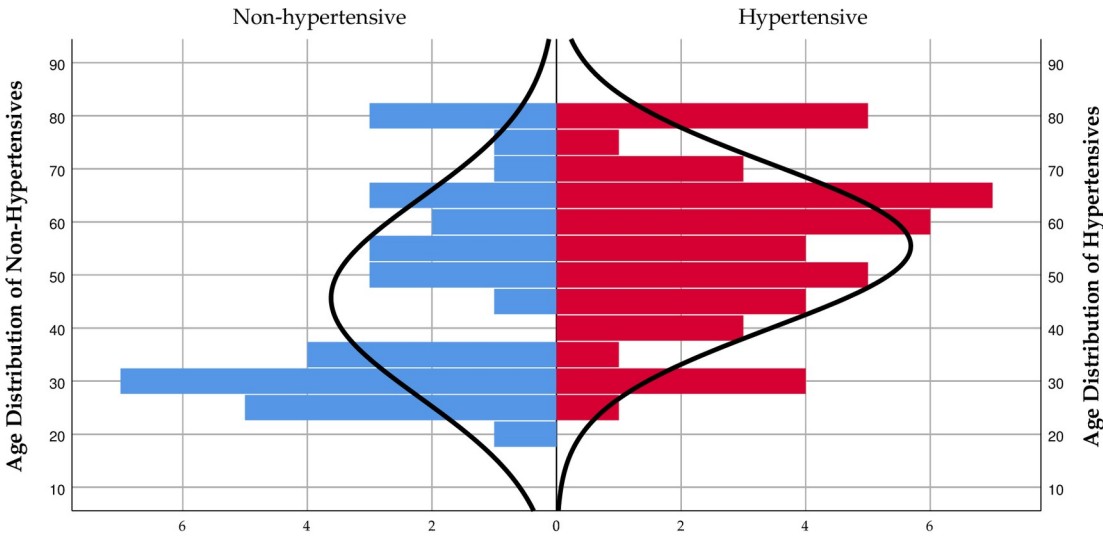

**Fig 3. Age distribution population pyramid over hypertension among male residents of Nyive.**

literature on the burden, treatment patterns, and the critical age predicting hypertension among rural folks in the Volta Region of Ghana. Hence, we aimed to fill this knowledge gap by designing the current study. Among a total of 417 rural community dwellers recruited into this study, we observed that the majority [402 (96.40%)] had been tested for hypertension, with more than half presenting with hypertension [212(50.84%)]. The rates of hypertension were 49.56% (168 participants) among females and 56.41% (44 participants) among males, with [137(32.85%)] previously diagnosed hypertension. The variation in the burden of rural hypertension (persons living with hypertension in rural areas) across various geographical areas are said to be attributed to the heterogeneity of hypertension studies including varying study methodologies, measurement techniques, study settings, the definition of hypertension,

**Table 5. Binary logistic age prediction of hypertension and critical age cutoff for hypertension among residents of Nyive.**

| Parameter | Total | Female | Male |
|---|---|---|---|
| Exp(B)(Odds Ratio) | 1.044 | 1.047 | 1.031 |
| Lower 95% CI of Exp(B) | 1.029 | 1.030 | 1.002 |
| Upper 95% CI of Exp(B) | 1.059 | 1.065 | 1.060 |
| Significance | <0.0001 | <0.0001 | 0.016 |
| **Percentage Correct Classification by Model** | | | |
| Non-Hypertension | 60.49 | 60.82 | 50.00 |
| Hypertension | 60.85 | 60.12 | 77.27 |
| Overall | 60.67 | 60.47 | 65.3 |
| **Age Predictive Threshold** | | | |
| Associated criterion | >39 | >39 | >35 |
| ROC curve (AUC) | 0.676 | 0.679 | 0.658 |
| Sensitivity | 87.74 | 88.69 | 86.36 |
| Specificity | 41.46 | 39.77 | 50.00 |
| p value | <0.0001 | <0.0001 | 0.0153 |

CI: Confidence Interval, ROC: Receiver Operating Characteristic, AUC: Area Under the Curve.

time study was conducted and the presence of hypertension risk factors [5, 7, 24]. The current finding, however, contributes to the growing evidence of increasing hypertension burden among Ghanaian rural dwellers over the years. Estimates from previous studies between 2004 and 2017 ranged from 24.1% to 44.7% [5–7, 10, 11]. In other studies, rural hypertension rates were reported to range from 32.3% in India [25] to 43% in Nigeria [26].

The crude prevalence of hypertension among the study population was estimated at 508.39 per 1000 residents. After age standardization, the male preponderance to hypertension observed in the current study was in line with an earlier report by Tuoyire and Ayetey [27] in the Ghana demographic and health survey, but opposite to that reported by Agyemang, Nyaaba [8] in the RODAM study. It is not quite apparent from this study what could be responsible for the male preponderance to hypertension. However, Bello [28] suggested that the phenomenon could be due to the high risk assumed by rural men owing to higher rates of alcohol and tobacco usage, as well as their socio-economic responsibility in providing for the family's upkeep. There are also assertions of male predilection for cardiovascular morbidity in middle-age which is mitigated in advancing years [29, 30].

We observed significantly increasing numbers of participants with hypertension who clustered at advanced ages in the hypertension group while a greater proportion of normotensives clustered at lower ages (Figs 1–3). A direct association between age and hypertension prevalence exists, often attributable to age-related structural changes in blood vessels potentially causing narrowing of the vascular lumen, and consequently increasing blood pressure [31–34]. Notably, the crude and age-standardized prevalence of age-group specific hypertension rose initially from 20 to 49 years before declining to the 70–79 years group (Table 4). The decline in the hypertension prevalence among the older age groups could be due to the phenomenon of excess mortality among the elderly with hypertension, a view Abdulle [35] held from a previous study. Moreover, the finding of a higher prevalence of hypertension among females, particularly those within the 60–79 years age categories compared to the menopausal age of 49 years could be explained by the effect of reduced oestrogen levels known to potentiate hypertension in advancing years [36, 37].

In this study, we found the risk of hypertension to increase by 4.4% (2.9%-5.9% at 95% CI) in the study population; a higher risk among females [4.7% (3.0%-6.5% at 95% CI)] compared to males [3.1% (0.2%-6.0% at 95% CI)] for a one year increase in age. The critical age of developing hypertension among the rural dwellers was >39 years, with a discriminating power of 0.679, similar to what was observed among females but lower in males (critical risk age was >35; discriminating power of 0.658) (Table 5). The observed age predicting hypertension (>39 years) in our study could suggest an early onset of hypertension among rural dwellers in Nyive; the onset was found to be even earlier among the male sub-population (>35 years). A similar age threshold for hypertension diagnosis was previously reported among urban dwellers in Kumasi and other parts of the country suggesting that the phenomenon is not limited to only rural dwellers in Nyive [12, 13, 38–40]. The earlier on-set of hypertension among men in this study may be explained by the lower oestrogen levels [37], but increased activities of the sympathetic nervous system and endothelin-1 leading to increased vasoconstriction and high blood pressure levels [41] compared to premenopausal women.

Standard treatment guidelines for hypertension recommend early diagnosis and commencement of treatment, in addition to making certain lifestyle changes to achieve optimal blood pressure control and prevent complications [42, 43]. Although pills and injectables are forms of antihypertensive medications widely in use [8], there are reports of patronage for alternative management with traditional healing and herbal preparations in Sub-Saharan Africa [44]. At the time of this study, 27(19.71%) previously diagnosed hypertensives were using herbal medications, while 17(12.41%) combined orthodox and herbal treatments

(Table 2). Though the choice of herbal medication did not significantly differ by demographic strata among the population, people with no education who also formed the majority of workers in the informal sector had higher percentage patronage of herbal medication (S1 Table). In the RODAM study, the age-standardized hypertension treatment for Ghanaian rural men and women was found to be 19% and 32% respectively [8]. In our study, the proportion of hypertensives with a preference to both orthodox and herbal treatments is, however, comparable to the previous reports of Kretchy, Sarkodie [45] in Accra and Kumasi (19.5%) and Ameade, Ibrahim [46] in Tamale (17.9%). While it is unclear why people would prefer alternative medication to orthodox treatment, or even combine both forms of treatment as observed in sub-Saharan Africa, the perceived failure of allopathic medications, traditional beliefs, health systems deficiencies, low socioeconomic status, and non-health insurance policy uptake was previously suggested as attributed factors [47, 48]. However, the simultaneous use of orthodox and herbal medicines can potentially result in therapeutic interactions leading to altered drug metabolism, exaggerated hypotensive effect, or decreased hypertension control with serious implications on the cardiovascular system and blood pressure control [49–52]. Approximately, only a third of previously diagnosed hypertensive participants [42 (30.66%)] had optimal blood pressure control, with similar proportions observed in both gender populations (male vs female; 35.00% vs 29.91%) (Table 2). The results suggest poor blood pressure control in the majority of hypertensives on treatment at the time of this study. Uncontrolled hypertension accounts for a greater proportion of stroke and heart failure cases among Ghanaians [53]. A wide range of factors have been proposed to contribute to the high rates of uncontrolled hypertension in less developed countries including lack of access to appropriate health care, cost, inadequate healthcare personnel to population ratio, and non-adherence to treatment and follow-up [24, 54, 55].

An important observation that equally merits attention is the high rate of prehypertension [96(34.29%)] and undiagnosed hypertension [75(26.79%)] among the rural dwellers. Among the newly diagnosed hypertensive participants, we observed 26(34.67%) presenting with stage two hypertension (Table 3). The results are, probably, the reaffirmation of the low awareness of hypertension status among populations in sub-Sahara Africa and rural dwellers in particular [56–58].

A potential limitation to the findings of this study is the inclusion of subjective definitions of hypertension and treatment patterns which could result from a recall bias on the part of the study participants.

## 5. Conclusion

Hypertension is high among rural adults in Nyive. The critical age at risk of hypertension and the estimated annual increased risk of hypertension were lower in males compared to females. High levels of undiagnosed hypertension and low levels of blood pressure control exist among the rural dwellers.

## Supporting information

**S1 Table. Treatment options among known hypertension residents of Nyive community stratified by demography.**
(DOCX)

**S1 Data.**
(SAV)

## Author Contributions

**Conceptualization:** James Osei-Yeboah, Ellis Owusu-Dabo, William K. B. A. Owiredu, Sylvester Yao Lokpo, Francis Delali Agode, Beatrice Bella Johnson.

**Data curation:** James Osei-Yeboah, Sylvester Yao Lokpo, Francis Delali Agode.

**Formal analysis:** James Osei-Yeboah, Sylvester Yao Lokpo, Francis Delali Agode.

**Funding acquisition:** James Osei-Yeboah, Ellis Owusu-Dabo, William K. B. A. Owiredu, Beatrice Bella Johnson.

**Investigation:** James Osei-Yeboah, Ellis Owusu-Dabo, Sylvester Yao Lokpo, Francis Delali Agode, Beatrice Bella Johnson.

**Methodology:** James Osei-Yeboah, Ellis Owusu-Dabo, William K. B. A. Owiredu, Sylvester Yao Lokpo, Francis Delali Agode, Beatrice Bella Johnson.

**Project administration:** James Osei-Yeboah, Ellis Owusu-Dabo, William K. B. A. Owiredu, Sylvester Yao Lokpo, Beatrice Bella Johnson.

**Resources:** James Osei-Yeboah, Ellis Owusu-Dabo, William K. B. A. Owiredu, Sylvester Yao Lokpo, Francis Delali Agode, Beatrice Bella Johnson.

**Supervision:** Ellis Owusu-Dabo, William K. B. A. Owiredu, Beatrice Bella Johnson.

**Validation:** James Osei-Yeboah, Ellis Owusu-Dabo, William K. B. A. Owiredu, Sylvester Yao Lokpo, Beatrice Bella Johnson.

**Visualization:** Ellis Owusu-Dabo, William K. B. A. Owiredu, Beatrice Bella Johnson.

**Writing – original draft:** James Osei-Yeboah, Sylvester Yao Lokpo, Francis Delali Agode.

**Writing – review & editing:** Ellis Owusu-Dabo, William K. B. A. Owiredu, Sylvester Yao Lokpo, Beatrice Bella Johnson.

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
