## [Decision Letter · Decision Letter 0]

14 Sep 2020

PONE-D-20-24599

The Community Burden of Hypertension and Treatment Patterns: An In-depth age predictor analysis: The Rural Community Risk of Non-Communicable Disease Study - (Nyive Phase I)

PLOS ONE

Dear Dr. Osei-Yeboah,

Thank you for submitting your manuscript to PLOS ONE. After careful consideration, we feel that it has merit but does not fully meet PLOS ONE’s publication criteria as it currently stands. Therefore, we invite you to submit a revised version of the manuscript that addresses the points raised during the review process.

I found this paper to be a potentially valuable addition to the literature.  Similar to the comments from the other reviewers, I believe that if you provide additional context for some of your statements, the paper will become more clear and understandable for readers who aren't familiar with the region you are studying in Ghana.  

Within the comments provided by the other reviewers, please ensure that you incorporate all the major or mandatory comments I've highlighted in parentheses.   We believe that once you incorporate these suggestions, the paper will be significantly improved and suitable for publication.  

We look forward to receiving your revised manuscript.

Kind regards,

Sonak D. Pastakia

Academic Editor

PLOS ONE

Additional Editor Comments:

The authors describe an interesting study describing their teams efforts to describe of hypertension amongst rural community members. My comments mirror the comments of the reviewers and would request that you address these comments to provide additional clarity on the issues they mention.

Journal Requirements:

5. Please include a copy of Table 6 which you refer to in your text (line 234).

Reviewers' comments:

Reviewer's Responses to Questions

**Comments to the Author**

1. Is the manuscript technically sound, and do the data support the conclusions?

Reviewer #1: Partly

Reviewer #2: Yes

2. Has the statistical analysis been performed appropriately and rigorously? 

Reviewer #1: I Don't Know

Reviewer #2: Yes

3. Have the authors made all data underlying the findings in their manuscript fully available?

Reviewer #1: Yes

Reviewer #2: Yes

4. Is the manuscript presented in an intelligible fashion and written in standard English?

Reviewer #1: Yes

Reviewer #2: Yes

5. Review Comments to the Author

Reviewer #1: Recommendation | The Community Burden of Hypertension and Treatment Patterns: An In-depth age

predictor analysis: The Rural Community Risk of Non-Communicable Disease Study -

(Nyive Phase I)

The manuscript talks about a health condition that is certainly of interest to healthcare and may contribute to literature. Hence, understanding risk factors to hypertension such as age and the effect of the condition on a community in addition to the "treatment patterns" available may help to improve care for hypertensive patients. 

This manuscript shows strengths in its introduction and clear lay out of the table of results. Authors random selection of participants is an additional strength.

Under discussion, it seems the authors are suggesting that the time period within which the study was done could be one of the factors contributing to the variation in the burden of hypertension. Not sure if I missed it but authors should consider indicating months/year data was collected to help readers connect to the above implication. Furthermore, the manuscript could benefit from an elaborate discussion on bridging the numbers from the results with the aims. What gap in literature will this fill?  (mandatory)

Under methods, authors should consider disclosing the number of all the participants contacted and those who opted out of the study. Did all eligible participants contacted by investigators consent to participate in the study? Kindly indicate the response rate and if the recorded number is small, consider indicating that as a limitation of the study. (mandatory if possible)

Authors should also consider describing the parts of the WHO STEPwise approach used since the study does not report on all the three steps. For example, There are other variables worth mentioning such as diet, smoking and exercise that affect hypertension but were not reported as part of the data collected. Did the questionnaire include all the variables in the WHO STEPwise approach? Consider stating this as potential limitation.  (optional)

Again, it is indicated in the manuscript that there were objective and subjective measures of hypertension by definition. The subjective report could also be a potential limitation.

"Rural hypertension" may not be a term that is widely known. Therefore, it will be appropriate to define what the term mean in the study at first mention. That is, please define "rural hypertension".  (Please make this more clear)

Other consideration:

Title: Consider revising the title, for instance, "Community Burden of Hypertension and Treatment Patterns - The Rural Community Risk of Non-Communicable Disease Study (Nyive Phase I): An In-depth age predictor analysis"

Abstract:

* Under methods; Sentence 2: Suggestion, "WHO STEPS Instrument... OR ..."The WHO STEPwise approach to noncommunicable disease risk factor surveillance (STEPS) instrument...." as stated on the WHO website instead of "The WHO STEPS wise approach for non-communicable diseases risk (NCD) instrument..." (minor)

Introduction:

* Review of literature seems fair although authors could consider information from more recent literature. Generally the review includes literature from Ghana and Sub-Saharan Africa. Almost half of the reference is 10 or more years. I suggest adding current literature such as: (major)

Methods:

*Line 33 & 34; revise the wording for the WHO instrument used as suggested under "Abstract" above.

* Line 100; Consider revising, "...is was the tool used to the collect data and ..."

* Line 115; Consider revising, "...Committee (JNC VII) on Prevention, Detection,..." NOT line 116 "...(JNC VII) criteria for blood pressure..."

*If data was collected before JNC VIII became available, then authors can ignore this recommendation. If not, JNC VIII guidelines have been available since December 2013. Consider adding a reason why JNC VII was preferred as reference for this study instead of the recent JNC VIII guidelines.

Results/Findings:

*Line 157, 162, 163 and 165; Consider revising, "... ever...". Are you trying to state "never"?

Discussion:

*Line 241; Consider revising, "...ever..."

*Line 243, Suggest, ".This rate of hypertension was...." and Line 244 to 245, "...with previously diagnosed hypertension."

*Line 296, Optimal BP control may differ based on individual patient condition and BP goal. Consider adding more information to clarify this assertion from the referenced literature.

Thank you for the opportunity to review this manuscript.

Reviewer #2: This paper is a significant contribution to the under-studied field of hypertension (HTN) control in Ghana. Although I recommend some significant revisions for clarity and context, I think it is an important analysis that merits publication. My comments by section follow below.

Introduction

There are a few typos and style issues to address - for instance the word "indicates" in line 53, and the use of "Africa" in line 55 when I presume "sub-Saharan Africa" would be more accurate. More broadly, however - the section could incorporate some additional context (space permitting) to make the paper's contribution clearer and stronger. For instance, the section makes reference to the Community Based Health Planning and Services (CHPS) program without citing or explaining it - consider detailing how this program works via, say, Phillips 2006 or other references. Similarly, the paper does not reference recent efforts to incorporate HTN care into CHPS, such as per Ogedegbe, 2018 and the TASSH study, or Haykin, 2020. There is also some limited data on HTN in Ghana in rural areas worth noting (e.g. Gomez-Olive, 2017). Detailing further what we currently know about attempts to measure and address the rural HTN burden in Ghana would make it easier for the reader to see why this new paper is noteworthy. (mandatory)

Methods

The study area section might benefit from a map, so persons unfamiliar with Ghana's geography can appreciate where the site is. Further explaining the features of the LLIN dataset would also help orient the reader, as would some more details on exactly how the current work builds on the Rural Community Risk of Non-Communicable Disease Study. (mandatory)

It is laudable that the study ensured that blood pressure was measured only after 3-5 minutes' rest, in a seated position, and documented this fact. Ideally, BP should be measured at arm's level, and the subject told not to eat or hold their urine prior to the BP check. Was this done? (mandatory)

The definitions of HTN use JNC 7 criteria rather than the 2017 ACC/AHA criteria. If possible, I would prefer the analysis be done using these cutoffs (if only as a sensitivity analysis), as they better reflect an updated understanding of the risk of elevated BP between 120 and 139 mm Hg systolic.  (optional)

Results

These are generally clear and straightforward, with a few typos (e.g., divorce vs divorced in table 1, and "having been" diagnosed with HTN in lines 163-164). I would suggest however rewording the paragraph in lines 162-171 for readability - I found it hard to follow what relative fraction of patients were using orthodox versus herbal medications versus both, as a proportion of all persons on treatment. These results are shown in table 2 but remain a bit confusing to me: for instance, how the bottom 3 rows (adding up to 73 persons) overlap with those treated in the last 2 weeks (63 persons) - were the other 10 persons off medicine for the last two weeks? I'd suggest making clearer the denominators (e.g., the 73 persons on treatment) relative to the numerators and percentages (e.g., 10, 46, and 17 persons on herbal versus orthodox versus both treatments).  (optional)

I'd also be keen to see if age, education level, and occupation (if not marital status) corresponds with tendency to take herbal versus non-herbal (orthodox) medication - the results are displayed only as a function of gender. Can this analysis also be added?  (optional)

Lastly - the finding that >35 and >39 are important age thresholds for HTN in men and women respectively is notable. However I was unable to follow the discriminating power calculation, and this does not seem to be discussed in the methods section. Can you kindly clarify?  (please clarify)

Discussion

This section is generally well-written, though I would be interested to hear more about the findings regarding herbal medication use (especially if we have more data as above on predictors of same). Are there other findings on this question besides Liwa and Smart's papers (references 36-38 - one is listed twice?) How much do these findings focus on Ghana and/or rural areas in particular? (optional)

I would be happy to review a revised version.

6. PLOS authors have the option to publish the peer review history of their article (what does this mean?). If published, this will include your full peer review and any attached files.

Reviewer #1: No

Reviewer #2: **Yes: **David J. Heller MD MPH

---

## [Author Response · Author response to Decision Letter 0]

19 Mar 2021

Reviewer 1

Comment: 

Under discussion, it seems the authors are suggesting that the time period within which the study was done could be one of the factors contributing to the variation in the burden of hypertension. Not sure if I missed it but authors should consider indicating months/year data was collected to help readers connect to the above implication. 

Response: 

Authors have provided the time period within which the study was carried out. We have included it in the methodology which can be found in the statement in lines 108-109 of the revised version of the manuscript.

Comment:

Furthermore, the manuscript could benefit from an elaborate discussion on bridging the numbers from the results with the aims. What gap in literature will this fill?

Response:

Authors have included statements in the discussion as suggested.

Comment:

Under methods, authors should consider disclosing the number of all the participants contacted and those who opted out of the study. Did all eligible participants contacted by investigators consent to participate in the study? Kindly indicate the response rate and if the recorded number is small, consider indicating that as a limitation of the study. (mandatory if possible).

Response:

Authors have included all information requested above by the reviewer and these can be traced to the statement in lines 118-120. However, we wish to state that there was an overwhelming acceptance rate of 98.12%, hence, this cannot be considered as a limitation for this study.

Comment:

Authors should also consider describing the parts of the WHO STEPwise approach used since the study does not report on all the three steps. For example, There are other variables worth mentioning such as diet, smoking and exercise that affect hypertension but were not reported as part of the data collected. Did the questionnaire include all the variables in the WHO STEPwise approach? Consider stating this as potential limitation. (optional)

Response:

Authors have included a detailed description of the instrument as suggested by the reviewer. The changes to this manuscript can be found in statements in lines 133-140. However, since the focus of this paper is clearly spelt out (to determine hypertension burden, treatment patterns and possible age predicting hypertension), it is the considered opinion of the authors that the exclusion of other parameters in the instrument for this study cannot be stated as a limitation.

Comment:

Again, it is indicated in the manuscript that there were objective and subjective measures of hypertension by definition. The subjective report could also be a potential limitation.

Response:

Authors have stated that the subjective measurement of hypertension included in this study is a potential limitation. The addition of this statement to the manuscript can be found in statements in lines 382-384.

Comment:

"Rural hypertension" may not be a term that is widely known. Therefore, it will be appropriate to define what the term mean in the study at first mention. That is, please define "rural hypertension". (Please make this more clear).

Response:

Authors have included the definition of “rural hypertension” at first mention in the manuscript as suggested by the reviewer. The change can be found in lines 285-286

Other Considerations

Comment:

Title: Consider revising the title, for instance, "Community Burden of Hypertension and Treatment Patterns - The Rural Community Risk of Non-Communicable Disease Study (Nyive Phase I): An In-depth age predictor analysis"

Response:

Authors have amended the title.

Comment:

Abstract:

* Under methods; Sentence 2: Suggestion, "WHO STEPS Instrument... OR ..."The WHO STEPwise approach to noncommunicable disease risk factor surveillance (STEPS) instrument...." as stated on the WHO website instead of "The WHO STEPS wise approach for non-communicable diseases risk (NCD) instrument..." (minor).

Response:

Authors have amended the sentence to reflect the reviewer’s suggestion

Comment:

Introduction:

* Review of literature seems fair although authors could consider information from more recent literature. Generally the review includes literature from Ghana and Sub-Saharan Africa. Almost half of the reference is 10 or more years. I suggest adding current literature such as: (major)

Response:

While every effort was made to include citations from more recent literature, authors wish to maintain the references- Cappuccio et al. (2004) and Bosu, 2010 for purposes of different time point analyses or comparisons.

Comment:

Methods:

*Line 33 & 34; revise the wording for the WHO instrument used as suggested under "Abstract" above.

* Line 100; Consider revising, "...is was the tool used to the collect data and ..."

Response:

Authors have revised the wordings as suggested.

Comment:

Line 115; Consider revising, "...Committee (JNC VII) on Prevention, Detection," NOT line 116 "...(JNC VII) criteria for blood pressure..."

*If data was collected before JNC VIII became available, then authors can ignore this recommendation. If not, JNC VIII guidelines have been available since December 2013. Consider adding a reason why JNC VII was preferred as reference for this study instead of the recent JNC VIII guidelines.

Response:

The definition of hypertension as used in the current analysis were based on the blood pressure cut-offs for different hypertension stages, this did not differ in the updated guideline, which is also the cut-off in use in the setting. 

Comment:

Results/Findings:

*Line 157, 162, 163 and 165; Consider revising, "... ever...". Are you trying to state "never"?

Response:

Author wish that the word “ever” is maintained in lines 157, 162, 163 and 165 since changing it to “never” as suggested will change the meaning of the statements where they appear

Comment:

Discussion:

*Line 241; Consider revising, "...ever..."

*Line 243, Suggest, ".This rate of hypertension was...." and Line 244 to 245, "...with previously diagnosed hypertension."

*Line 296, Optimal BP control may differ based on individual patient condition and BP goal. Consider adding more information to clarify this assertion from the referenced literature.

Response:

Authors have made changes to statements as suggested by the reviewer in lines 283-285.

Concerning issues in line 296 in the previous version of the manuscript, authors addressed the issues by taking out the blood pressure cut-offs “140/90” to make the statement more appropriate

Reviewer 2

Comment:

Introduction

There are a few typos and style issues to address - for instance the word "indicates" in line 53, and the use of "Africa" in line 55 when I presume "sub-Saharan Africa" would be more accurate. More broadly, however - the section could incorporate some additional context (space permitting) to make the paper's contribution clearer and stronger. For instance, the section makes reference to the Community Based Health Planning and Services (CHPS) program without citing or explaining it - consider detailing how this program works via, say, Phillips 2006 or other references. Similarly, the paper does not reference recent efforts to incorporate HTN care into CHPS, such as per Ogedegbe, 2018 and the TASSH study, or Haykin, 2020. There is also some limited data on HTN in Ghana in rural areas worth noting (e.g. Gomez-Olive, 2017). Detailing further what we currently know about attempts to measure and address the rural HTN burden in Ghana would make it easier for the reader to see why this new paper is noteworthy. (mandatory)

Response:

Authors have addressed the typo and style issues. We have also included additional information on the CHPS program explaining how attempts have been made to incorporate hypertension care into CHPS to address the issue of rural hypertension and the associated challenges. The changes made as suggested by the reviewer can be traced to statements in lines 75-89.

Comment:

The study area section might benefit from a map, so persons unfamiliar with Ghana's geography can appreciate where the site is. Further explaining the features of the LLIN dataset would also help orient the reader, as would some more details on exactly how the current work builds on the Rural Community Risk of Non-Communicable Disease Study. (mandatory).

Response:

Authors have included the map and the explanation of the LLIN based dataset which was used to determine the projected population as suggested. 

Comment:

Ideally, BP should be measured at arm's level, and the subject told not to eat or hold their urine prior to the BP check. Was this done? (mandatory).

Response:

Participants’ blood pressure was measured at an arm’s level during the time they were fasting and this statement have been included in the statement in lines 144-145.

Comment:

The definitions of HTN use JNC 7 criteria rather than the 2017 ACC/AHA criteria. If possible, I would prefer the analysis be done using these cutoffs (if only as a sensitivity analysis), as they better reflect an updated understanding of the risk of elevated BP between 120 and 139 mm Hg systolic. (optional).

Response:

Authors adopted the JNC 7 criteria for hypertension definition because the criteria was still in use by the Ghana Health Service in the diagnosis and management of hypertension.

Comment:

These are generally clear and straightforward, with a few typos (e.g., divorce vs divorced in table 1, and "having been" diagnosed with HTN in lines 163-164). I would suggest however rewording the paragraph in lines 162-171 for readability - I found it hard to follow what relative fraction of patients were using orthodox versus herbal medications versus both, as a proportion of all persons on treatment. These results are shown in table 2 but remain a bit confusing to me: for instance, how the bottom 3 rows (adding up to 73 persons) overlap with those treated in the last 2 weeks (63 persons) - were the other 10 persons off medicine for the last two weeks? I'd suggest making clearer the denominators (e.g., the 73 persons on treatment) relative to the numerators and percentages (e.g., 10, 46, and 17 persons on herbal versus orthodox versus both treatments). (optional).

Response:

The scheme of analysis in table 2, was to profile or tell a story among participants who knew they were hypertensive and evaluate their practices (thus questions were answerable only when you answer yes of ever being diagnosed of hypertension). Since it obvious these questions would not be applicable to a non-hypertensive person, the denominator was 137 for the total, 117 for female and 20 for the male (representing self-reported known hypertensives).

Comment:

I'd also be keen to see if age, education level, and occupation (if not marital status) corresponds with tendency to take herbal versus non-herbal (orthodox) medication - the results are displayed only as a function of gender. Can this analysis also be added? (optional).

Response:

Authors appreciate the possible additional information in describing the treatment pattern as a function of age, educational attainment and occupation, we have added that part of the analysis as a supplementary information in Table S1.

Comment:

Lastly - the finding that >35 and >39 are important age thresholds for HTN in men and women respectively is notable. However I was unable to follow the discriminating power calculation, and this does not seem to be discussed in the methods section. Can you kindly clarify? (please clarify).

Response:

The method used for determining the discriminating power of the age cutoff points [the area under the curve (AUC) of the receiver -operator characteristic (ROC)] was stated in the materials and methods under the statistical analysis section.

Comment:

Discussion

This section is generally well-written, though I would be interested to hear more about the findings regarding herbal medication use (especially if we have more data as above on predictors of same). Are there other findings on this question besides Liwa and Smart's papers (references 36-38 - one is listed twice?) How much do these findings focus on Ghana and/or rural areas in particular? (optional)

Response:

Authors have included local findings in the discussion as suggested.

---

## [Decision Letter · Decision Letter 1]

14 May 2021

Community Burden of Hypertension and Treatment Patterns: An In-depth age predictor analysis: (The Rural Community Risk of Non-Communicable Disease Study - (Nyive Phase I)

PONE-D-20-24599R1

Dear Dr. Osei-Yeboah,

We’re pleased to inform you that your manuscript has been judged scientifically suitable for publication and will be formally accepted for publication once it meets all outstanding technical requirements.

Kind regards,

Sonak D. Pastakia

Academic Editor

PLOS ONE

Additional Editor Comments (optional):

This version is considerable improvement from the last version. I would correct a couple of the editorial typos mentioned by the reviewers prior to publication.

Reviewers' comments:

Reviewer's Responses to Questions

**Comments to the Author**

1. If the authors have adequately addressed your comments raised in a previous round of review and you feel that this manuscript is now acceptable for publication, you may indicate that here to bypass the “Comments to the Author” section, enter your conflict of interest statement in the “Confidential to Editor” section, and submit your "Accept" recommendation.

Reviewer #1: (No Response)

Reviewer #2: (No Response)

2. Is the manuscript technically sound, and do the data support the conclusions?

Reviewer #1: Yes

Reviewer #2: Yes

3. Has the statistical analysis been performed appropriately and rigorously? 

Reviewer #1: Yes

Reviewer #2: Yes

4. Have the authors made all data underlying the findings in their manuscript fully available?

Reviewer #1: Yes

Reviewer #2: Yes

5. Is the manuscript presented in an intelligible fashion and written in standard English?

Reviewer #1: Yes

Reviewer #2: Yes

6. Review Comments to the Author

Reviewer #1: Thanks again for the opportunity to review this manuscript and for considering reviewers suggestions.

Overall, great improvement in clarity, detail and cohesiveness. I recognize the attempt to connect the findings in this study to previous study, however, Doumas et al., 2013 does not seem to fit the statement made (page 24, line 334-337). The need for cardio-protection for premenopausal women on certain therapies may be beneficial but the point on how that can help explain early on-set of hypertension in men is not clear? Please clarify or elaborate so that readers understand how the statement supports the claim.

Reviewer #2: I am satisfied by the thoughtful revisions provided by the authors. These include adding key context to the introduction section; a helpful map of the target area; and clarifying how blood pressure was checked.

I am fine with using the JNC-7 guidelines given these are still used by the Ghana Health Service. I also found the clarifications helpful (in the table and the text) regarding orthodox versus herbal medication.

I propose the paper be accepted essentially as-is, but I propose some minor edits beforehand:

1. Ideally please add a reference in/around line 152 indicating that JNC7 remains the hypertension guideline of choice for the Ghana Health Service;

2. Please indicate in line 205 that the numbers for the three categories of medications used (orthodox, herbal, or both) add up to 53.28% (the total who are on medication)

3. Please proof the paper once more for typos that spell check might have missed (e.g., thumb instead of tomb, line 188).

4. Lastly, forgive any methodological ignorance on my part, but I don’t quite understand what is meant (at a practical level) by the critical at risk age of 35 for men and 29 for women. Can the authors add a line explaining expressly what this means for the clinician?

I have no further comments and thank the reviewers for their edits.

7. PLOS authors have the option to publish the peer review history of their article (what does this mean?). If published, this will include your full peer review and any attached files.

Reviewer #1: **Yes: **Akua A. Appiah-Num, PharmD, MS

Reviewer #2: No

---

## [Editor Report · Acceptance letter]

31 May 2021

PONE-D-20-24599R1 

Community Burden of Hypertension and Treatment Patterns: An In-depth age predictor analysis: (The Rural Community Risk of Non-Communicable Disease Study - Nyive Phase I) 

Dear Dr. Osei-Yeboah:

I'm pleased to inform you that your manuscript has been deemed suitable for publication in PLOS ONE. Congratulations! Your manuscript is now with our production department. 

Kind regards, 

on behalf of

Dr. Sonak D. Pastakia 

Academic Editor

PLOS ONE